# DNA Methylation of the Dopamine Transporter *DAT1* Gene—Bliss Seekers in the Light of Epigenetics

**DOI:** 10.3390/ijms24065265

**Published:** 2023-03-09

**Authors:** Krzysztof Chmielowiec, Jolanta Chmielowiec, Jolanta Masiak, Aleksandra Strońska-Pluta, Milena Lachowicz, Agnieszka Boroń, Dariusz Larysz, Magdalena Dzitkowska-Zabielska, Paweł Cięszczyk, Anna Grzywacz

**Affiliations:** 1Department of Hygiene and Epidemiology, Collegium Medicum, University of Zielona Góra, 28 Zyty St., 65-046 Zielona Góra, Poland; 2Second Department of Psychiatry and Psychiatric Rehabilitation, Medical University of Lublin, 1 Głuska St., 20-059 Lublin, Poland; 3Independent Laboratory of Health Promotion, Pomeranian Medical University in Szczecin, Aleja Powstańców Wielkopolskich 72 St., 70-111 Szczecin, Poland; 4Department of Psychology, Gdansk University of Physical Education and Sport, 80-336 Gdansk, Poland; 5Department of Clinical and Molecular Biochemistry, Pomeranian Medical University in Szczecin, Aleja Powstańców Wielkopolskich 72 St., 70-111 Szczecin, Poland; 6109 Military Hospital with Cutpatient Cinic in Szczecin, Piotra Skargi 9-11 St., 70-965 Szczecin, Poland; 7Faculty of Physical Education, Gdansk University of Physical Education and Sport, 80-336 Gdansk, Poland

**Keywords:** DAT1 gene, promoter methylation, dopamine, nicotine addiction, addiction, athletes

## Abstract

DNA methylation (leading to gene silencing) is one of the best-studied epigenetic mechanisms. It is also essential in regulating the dynamics of dopamine release in the synaptic cleft. This regulation relates to the expression of the dopamine transporter gene (*DAT1*). We examined 137 people addicted to nicotine, 274 addicted subjects, 105 sports subjects and 290 people from the control group. After applying the Bonferroni correction, our results show that as many as 24 out of 33 examined CpG islands had statistically significantly higher methylation in the nicotine-dependent subjects and athletes groups compared to the control group. Analysis of total *DAT1* methylation revealed a statistically significant increase in the number of total methylated CpG islands in addicted subjects (40.94%), nicotine-dependent subjects (62.84%) and sports subjects (65.71%) compared to controls (42.36%). The analysis of the methylation status of individual CpG sites revealed a new direction of research on the biological aspects of regulating dopamine release in people addicted to nicotine, people practicing sports and people addicted to psychoactive substances.

## 1. Introduction

DNA methylation is a post-replication process in which a methyl group (-CH3) is added to the cytosine in the appropriate sequence context. DNA methylation (leading to gene silencing) is one of the best-studied epigenetic mechanisms. This is often done within the CpG islands, where cytosine conversion to 5-methyl-cytosine blocks gene expression [1]. This causes gene silencing by several mechanisms. One of these mechanisms is direct interference with the DNA binding of specific transcription factors, such as AP-2, c-Myc/Myn, cyclic AMP-dependent activator CREB, E2F and NF-κB, which recognize sequences containing CpG residues [2]. The CpG-rich regions overlap with the promoters of 50–60% of human genes and are usually more methylated than the CpG dinucleotides found outside the islands [3,4,5].

The DNA methylation pattern is stable through cell division and somatically inherited. It can be formed by de novo methylation or by demethylation. DNA methylation is a unique way to encode information in a stable but reversible way [1].

Correct control of DNA methylation is essential for proper cell function. DNA methylation dysfunction has been linked to many human pathologies [6,7,8,9].

The presented project aims to determine epigenetic factors which may significantly influence operational differences in the individual regulation of the dopaminergic system related to the different pathways of dopamine arousing. We chose three groups of people who, at first glance, are entirely unrelated: addicts to psychoactive substances, addicts to nicotine and professional athletes. In the first of them, an addictive substance is a source of dopaminergic reward, while in the second one, the natural forms of stimulation, i.e., sport or competition, result in a dopamine feast, and the postsynaptic dopaminergic receptors furiously demand a new dose to supplement the dopamine, which has not been effective enough so far.

Traditional epigenetic studies emphasize the importance of the complex interplay between the *DAT1* gene and the environment [10]. The dopamine transporter *DAT1* is crucial in dopamine (DA) neurotransmission. It is a protein located on nerve endings that modulates the dynamics and levels of dopamine release (by recycling extracellular dopamine back to the presynaptic terminal, thus terminating its action). Dysregulated DA activity may be due to altered release or reuptake; therefore, properly regulating *DAT1* gene expression is crucial for maintaining homeostasis in dopaminergic systems.

The human *DAT1* gene is extremely GC-dense and has multiple CpG islands, unlike most vertebrate genes, which only have a CpG island overlapping the pro-motor. The inactive promoters of the human cytosine and guanine genes account for an average of 57% of nucleotides [11], but GCs account for 79% of the DAT1 promoter sequence. The most notable feature of human *DAT1* is its high sensitivity to epigenetic regulation: in contrast to the relative enrichment of GC nucleotides in the proximal promoter region, as is the case in most human genes, the entire *DAT1* locus has a sequence composition of GC-bias (0, 55) and includes multiple CpG sites comprising 27 bona fide CGIs (CpG islands). The epigenetic sensitivity of the *DAT1* gene has increased throughout evolution. The genetic drift of the GC nucleotide accumulation-oriented *DAT1* sequence may reflect its amplifying epigenetic potential, which is essential in the regulatory processes resulting from the increasingly complex functions of the human brain [12].

The dopaminergic system plays a significant role in amplifying the effects of drug abuse [13,14]. Three factors contribute to the development of addiction: genetic, environmental diversity and the effect of drugs on gene expression or mRNA levels [15]. Recent studies have shown that transcription factors such as non-coding RNAs, histone modifications and chromatin structures can alter the transcription potential of genes. Transcription factors also contribute significantly to many neural adaptations resulting from chronic drug exposure [16]. Growing evidence supports the hypothesis that drug abuse directly influences the mechanism of epigenetic regulation and that these adaptations are among the main processes by which drugs induce highly stable changes in the brain that mediate the addiction phenotype [17]. Important epigenetic mechanisms include DNA methylation, histone modifications and small non-coding RNAs [18,19].

Cigarette smoke is one of DNA methylation’s most potent environmental modifiers [20]. Cigarette smoke can modulate it through DNA damage followed by DNMT recruitment. DNA repair sites recruit DNMT1 [21], which methylates the CpG adjacent to the repaired nucleotides [22]. Cigarette smoke may also modulate DNA methylation through nicotine’s effect on gene expression [23].

DNA methylation is stable through cell division dynamic throughout life due to environmental stimuli, such as exercise and nutrition. Exercise epigenetics is a new and exciting field of research, and we currently have limited knowledge of whether and how epigenetic signals, such as DNA methylation, mediate responses to exercise. A study from 2012 showed lower DNA methylation in specific genes 20 min after high-intensity endurance exercise and demonstrated rapid dynamics of DNA methylation. A meta-analysis of 16 studies identified 478 loci (307 in skeletal muscle) that undergo methylation changes following either acute (one bout) or regular exercise (walking, cycling, and tai-chi). Additional studies have found that long-term activity is associated with changes in DNA methylation in human skeletal muscles [24]. However, there is little research on epigenetics, particularly methylation in athletes. Researchers analyze various genes and their methylation levels in the promoter regions. We focus on that area of research based on the reports regarding the effects of dopamine on motivation and a series of thought processes. This is not the only study into the genetics and epigenetics of the gene regions responsible for physical exercise focusing on the endurance or functional phase.

Therefore, we want to examine how various factors, such as addiction to psychoactive substances, nicotine addiction or practicing sports, affect the methylation of the *DAT1* gene promoter.

## 2. Results

Analysis of the methylation status of individual CpG sites revealed differences in methylation levels at individual sites (islands) of the *DAT1* promoter (Table 1, Figure 1). At twenty-four of the 33 CpG sites, a significant methylation level was found in the addicted, nicotine-dependent and sports subjects (sites 1, 2,3, 4, 5, 7, 8, 9, 10, 11, 12, 13, 14, 15, 16, 17, 20, 21, 23, 24, 26, 27, 32 and 33).

In CpG islands 1, 2, 4, 12, 14, 15, 16, 20, and 24, the highest methylation was shown in sports, nicotine-dependent, control and addicted subjects, respectively. In CpG islands 3, 10, 13, 17, 21, and 27, the highest degree of methylation was shown in sports, nicotine-dependent, addicted and control subjects, respectively. In CpG islands 5, 7, 9, and 23, the highest degree of methylation was shown in nicotine-dependent, sports, control and addicted subjects, respectively. In CpG islands 8, 11, and 26, the highest degree of methylation was shown in the nicotine-dependent, sports, addicted and control subjects, respectively. In CpG island 32, the highest degree of methylation was demonstrated in the addicted, sports, control and nicotine-dependent subjects, respectively. In CpG island 33, the highest degree of methylation was shown in sports, addicted, control and nicotine-dependent subjects, respectively.

Analysis of total *DAT1* methylation revealed a statistically significant increase in the number of total methylated CpG islands in addicted (40.94%), nicotine-dependent (62.84%) and sports subjects (65.71%) compared to controls (42.36%) (F_3,802_ = 95.185, *p* < 0.000001, η^2^ = 0.262, Table 2, Figure 2). Table 3 shows the results of the post hoc test.

## 3. Discussion

The study of epigenetic factors, especially methylation in the promoter region of the *DAT1* gene, has been the subject of our interest for many years. The presented results culminate in analyzing groups commonly referred to as “dopamine seekers”, i.e., people using nicotine, people addicted to psychoactive substances and professional athletes. In our previous reports, we analyzed 33 CpG sites located in the promoter region of the *DAT1* gene in cannabis-dependent subjects and controls. The results of our analyses showed significant differences between both groups—methylation changes are not similar across all sites. Compared with the control group, some sites in the dependent subjects were hypermethylated, while some were hypomethylated. Further, the assessment of the ability of transcription factors to bind to indicated sites revealed a substantial number of those regulators of gene expression [25].

In another study, we analyzed 33 CpG sites of the promoter region of the *DAT1* gene among a sample group of athletes and a control group. The obtained data showed significant differences between the two groups as methylation changes across all sites differed. When comparing the control group and the group of athletes, the latter were hypermethylated, and some were hypomethylated. Moreover, transcription factors’ ability to bind to chosen sites revealed a great number of those regulators of gene expression [26].

In the present study, we went one step further by analyzing people addicted to various substances (not only marijuana), people using nicotine and athletes with championship successes. In our study of individual *DAT1* methylation CpG islands (1,2,4,9,10,11,12,13,14,15,16,17,21,24,26,27), a significantly higher percentage of methylation was shown in nicotine-dependent and sports subjects compared to addicted subjects and controls (Table 1, Figure 1). The total percentage of methylation also showed a statistically significantly higher percentage of methylation in nicotine-dependent and sports subjects compared to addicted subjects and control (Table 2 and Table 3, Figure 2) (62.84% and 65.71% vs. 40.94% and 42.36%, respectively)

The epigenetic regulation of gene expression can be mediated by both hypermethylation and hypomethylation of DNA.

Our research showed that in the group of addicts, the average total methylation level of the *DAT1* gene promoter was lower than in the control group (40.94% vs. 42.36%; Table 3), which indicates the occurrence of DNA hypomethylation.

Imaging studies in heroin addicts using SPECT have shown less DAT availability in the bilateral caudate nucleus, putamen and striatum. Unfortunately, the exact mechanism of changes in the density of the dopamine transporter is still unknown [27,28].

The lower availability of DAT indicates lower gene expression in the studied regions, which our current finding of reduced levels of DNA methylation, or hypomethylation, could explain.

However, in the group of people addicted to nicotine and the group of athletes, the dominant epigenetic phenomenon turned out to be the process of DNA hypermethylation. The average total methylation level of the *DAT1* gene in the group of people addicted to nicotine was significantly higher than in the control group (62.84% vs. 42.36%; Table 3).

Cigarette smoke may also modulate DNA methylation through nicotine’s effect on gene expression [23]. Cigarette smoke may indirectly affect changes in DNA methylation by modulating DNA expression and binding activity factors. For example, cigarette smoke condensate increased the expression of Sp1 and DNA binding in lung epithelial cells [29,30]. Sp1 is a general transcription factor that binds to GC-rich motifs in the promoters of genes [31]. Transcription factor Sp1 was shown to contribute to regulating DAT mRNA expression [32]

Abdelkhalek et al. measured methylation in the promoter IV of the BDNF gene in smokers compared with healthy controls within 14 days of smoking cessation. Mean methylation was significantly higher in smokers than in healthy controls at all time points [33].

Halit et al. assessed the relationship between methylation in the promoters of the APC, NR3C1, and DRD2 genes and its effect on nicotine use. They found a significant difference in APC2 methylation, comparing the study and control groups. Smokers’ methylation rate was about eight times higher than healthy controls. NR3C1 methylation was slightly higher in patients with nicotine addiction compared to controls, but the difference was not significant between the two groups (% 95.33 vs. 91.08, *p* = 0.269). The DRD2 methylation ratio was insignificant between nicotine-addicted patients and healthy controls (*p* = 0.894) [34].

These previously documented studies allow us to conclude that the increased expression of Sp1 due to tobacco smoke may also increase the intensity of the dopamine transporter gene transcription process, which was revealed in our study in the form of DAT promoter hypermethylation.

Moreover, cigarette smoke can change DNA methylation through hypoxia—cigarette smoke contains carbon monoxide, which binds to haemoglobin (competitively with oxygen), thus reducing the oxygenation of tissues [35]. Hypoxia leads to HIF-1α-dependent up-regulation of methionine adenosyltransferase 2A, an enzyme synthesizing S-adenosylmethionine, the leading biological methyl donor critical to DNA methylation processes [36].

The present study provides further evidence that changes in DNA methylation in the regulatory (promoter) regions of addiction-related genes are associated with nicotine addiction, possibly because they can result in altered gene transcription. The rationale is that promoter DNA methylation can directly interfere with binding transcription factors (TFs) to regulatory regions. We also proved hypermethylation of the DAT promoter in a group of athletes. The average total methylation level of the *DAT1* gene in the group of people practicing sports was significantly higher than in the control group (65.71% vs. 42.36%; Table 3). It is visible that sport is an essential environmental factor positively influencing the expression of the *DAT1* gene.

We must remember that DNA methylation patterns are susceptible to change throughout life. Environmental factors acting through DNA methylation can modify our health, both positively and negatively. Moreover, these changes can be passed on to the next generations, participating in the evolution process [37]. In our work, we proved the extraordinary effect of epigenetics on the dopaminergic system in three different groups of happiness seekers.

## 4. Materials and Methods

### 4.1. Participants

The study group consisted of three homogeneous subgroups: patients addicted to drugs (274 patients), people addicted to nicotine (137 people) and athletes (105 people). The control group consisted of 290 people who were not addicted to drugs or nicotine and were not athletes. The average age of the individual groups and the percentages of females and males are shown in Table 4. To reduce the possibility of genetic admixture and overcome potential problems from population stratification, all participants were European.

The study was approved by the bioethics committee of the Pomeranian Medical University in Szczecin and was conducted following the guidelines of the Declaration of Helsinki. All participants were informed about the rules of the study, familiarized with its course and told about the possibility of withdrawing from the study at any time.

None of the study participants was financially rewarded for participating in the project, and the study was wholly anonymized following the principles of personal data protection. The addicted subjects were recruited in addiction treatment facilities in the province of Lubuskie after at least 3 months of abstinence. Nicotine dependence was tested with the Fagerstrom test. Athletes have been defined as people who practice sports professionally. The control group was selected in terms of age and sex; people who had not smoked more than a pack of cigarettes in their lifetime and were not addicted to other psychoactive substances at the time of the examination were not athletes. All procedures for comfort and concentration were performed.

### 4.2. Assessment of the Methylation Status of the Dopamine Gene Transporter (DAT1) Promoter

As previously described, DNA was isolated from peripheral blood using a DNA isolation kit (A&A Biotechnology, Gdynia, Poland) [37]. Bisulfite modifications of 250 ng DNA were performed following the manufacturer’s instructions using the EZ DNA Methylation Kit (Zymo Research, Orange, CA, USA). The methylation-specific PCR assay was performed on a Mastercycler epgradient S (Eppendorf, Germany).

Oligonucleotide primers designed using metprimer (http://www.urogene.org/cgi-bin/methprimer/methprimer.cgi, accessed 29 April 2022) were obtained from Genomed.pl (Warsaw, Poland). The status of the *DAT1* promoter (ENSG00000142319) was assessed by PCR using primers specific for the gene fragment, i.e., DATF: 5’-GGTTTTTGTTTTTTTTATGTTGAG-3’; DATR: 5’-AAATCCCCTAAACCTAATCCC-3’. Table 5 shows the PCR conditions used to amplify the 447 bp fragment spanning the 33 CpG sites in the *DAT1* gene promoter.

The magnesium chloride ion concentration was 2.5 mM. After the amplification, the PCR products were sequenced as previously described [38]. Briefly, samples were verified by sequencing with the BigDye v3.1 kit (Applied Biosystems, Darmstadt, Germany) and separation by ethanol extraction with ABI Prism 3130XL (Applied Biosystems, Darmstadt, Germany) in a 36 cm POP7 Polymer capillary, using a reverse-primer.

Sequencing chromatograms were analyzed by the 4peaks software (v. 1.8., Mek & Tosj, Amsterdam, The Netherlands, https://nucleobytes.com/4peaks/index.html, accessed 29 April 2022). Cytosine methylation was considered positive when the G/A+G ratio was at least 20% of the total signal. The mathematical equation to calculate the percentage of methylation in each participant was: (G/(G + A) × 100)

### 4.3. Statistical Analysis

Analysis and comparison of the total methylation level (%) of 33 CpG *DAT1* sites in the four analyzed groups of test subjects were performed using the One-Way Analysis of Variance (addicted subjects; nicotine-dependent subjects; sports subjects; control)).

The chi-squared test was used to show differences in the percentage of methylation in individual CpG islands in the analyzed four groups of subjects, with *p* < 0.05 regarded as statistically significant. The Bonferroni, multiple comparisons correction was applied for these variables, and the accepted significance level was 0.0015 (0.05/33). All computations were performed using STATISTICA 13 (TIBCO Software, Inc., Palo Alto, CA, USA) and PQStat Software (v. 1.8.2., Poznań, Poland).

## 5. Conclusions

Our analysis of the methylation status of individual CpG sites revealed a new direction of research on the biological aspects of regulating dopamine release in people addicted to nicotine, people practicing sports and people addicted to psychoactive substances.

## Figures and Tables

**Figure 1 ijms-24-05265-f001:**
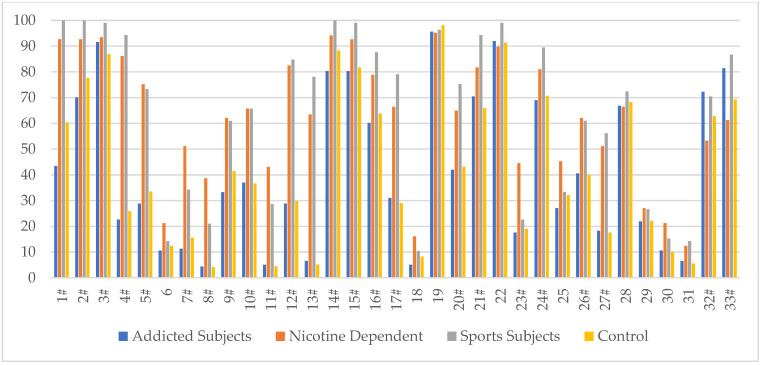
The methylation level (%) of 33 CpG *DAT1* sites in a group of addicted subjects, nicotine-dependent subjects, sports subjects and controls. # statistically significant differences in the methylation level—Bonferroni correction was applied, and *p*-value was lowered to 0.0015 (*p* = 0.05/33 (number of statistical tests performed)).

**Figure 2 ijms-24-05265-f002:**
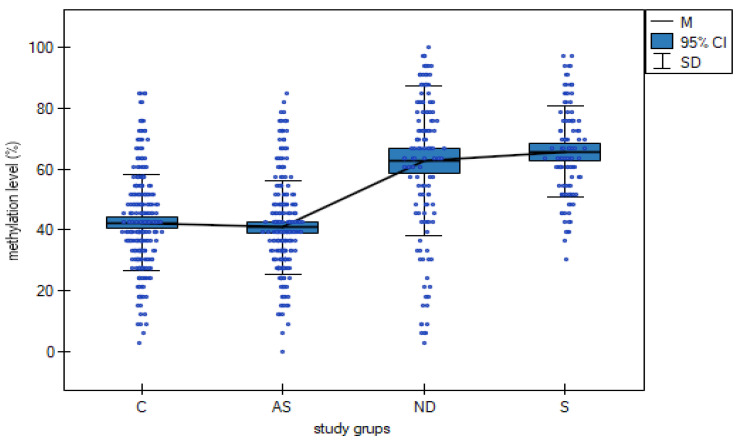
The relationship between total methylation sites (%) in a group of AS—addicted subjects, ND—nicotine-dependent subjects, S—sports subjects and C—controls. Mt—mean total methylation sites. SD—standard deviation.

**Table 1 ijms-24-05265-t001:** Methylation status of 33 CpG *DAT1* sites in the addicted, nicotine-dependent and sports subjects. 274 addicted, 137 nicotine-dependent, 105 sports and 290 control subjects were tested to compare the methylation status at the indicated CpG sites.

CpG Site	Study Groups Methylation Level n (%)				Test χ^2^	*p* Value
	Addicted Subjects n = 274	Nicotine Dependent n = 137	Sports Subjects n = 105	Control n = 290		
1#	119 43.43%	127 92.70%	105 100.00%	175 60.34%	162.079	0.0000
2#	192 70.07%	127 92.70%	105 100.00%	225 77.59%	59.020	0.0000
3#	251 91.61%	128 93.43%	104 99.05%	252 86.90%	15.640	0.0013
4#	62 22.63%	118 86.13%	99 94.29%	75 25.86%	296.076	0.0000
5#	79 28.83%	103 75.18%	77 73.33%	97 33.45%	129.303	0.0000
6	29 10.58%	29 21.17%	15 14.29%	36 12.41%	9.228	0.0264
7	31 11.31%	70 51.09%	36 34.29%	45 15.52%	100.119	0.0000
8#	12 4.38%	53 38.69%	22 20.95%	12 4.15%	129.448	0.0000
9#	91 33.21%	85 62.04%	64 60.95%	120 41.38%	43.821	0.0000
10#	101 37.00%	90 65.69%	69 65.71%	106 36.55%	57.162	0.0000
11#	14 5.11%	59 43.07%	30 28.57%	13 4.48%	150.831	0.0000
12#	79 28.83%	113 82.48%	89 84.76%	86 29.66%	200.864	0.0000
13#	18 6.57%	87 63.50%	82 78.10%	15 5.17%	376.015	0.0000
14#	220 80.29%	129 94.16%	105 100.00%	256 88.28%	34.897	0.0000
15#	220 80.29%	127 92.70%	104 99.05%	237 81.72%	30.348	0.0000
16#	165 60.22%	108 78.83%	92 87.62%	185 63.79%	36.067	0.0000
17#	85 31.02%	91 66.42%	83 79.05%	84 28.97%	125.943	0.0000
18	14 5.11%	22 16.06%	11 10.48%	24 8.28%	14.097	0.0027
19	262 95.62%	276 95.17%	132 96.35%	103 98.10%	1.805	0.6139
20#	115 41.97%	89 64.96%	79 75.24%	125 43.10%	51.489	0.0000
21#	193 70.44%	112 81.75%	99 94.29%	191 65.86%	38.342	0.0000
22	252 91.97%	123 89.78%	104 99.05%	265 91.38%	8.345	0.0394
23#	48 17.52%	61 44.53%	48 22.64%	55 18.97%	62.842	0.0000
24#	189 68.98%	111 81.02%	94 89.52%	205 70.69%	22.039	0.0001
25	74 27.01%	62 45.26%	35 33.33%	93 32.07%	13.907	0.0030
26#	111 40.51%	85 62.04%	64 60.95%	116 40.00%	30.977	0.0000
27#	50 18.25%	70 51.09%	59 56.19%	51 17.59%	104.836	0.0000
28	183 66.79%	91 66.42%	76 72.38%	198 68.28%	1.277	0.7345
29	60 21.90%	37 27.01%	28 26.67%	64 22.07%	2.247	0.5228
30	29 10.58%	29 21.17%	16 15.24%	29 10.00%	12.413	0.0061
31	18 6.57%	17 12.41%	15 14.29%	16 5.52%	12.146	0.0069
32#	198 72.26%	73 53.28%	74 70.48%	182 62.76%	16.676	0.0008
33#	223 81.39%	84 61.31%	91 86.67%	201 69.31%	31.511	0.0000

χ^2^ (*p*)—chi-square test (significance level); #—differences in the level of methylation at the limit of statistical significance; n—number of subjects. # Bonferroni correction was applied, and the *p*-value was lowered to 0.0015 (*p* = 0.05/33 (number of statistical tests performed)).

**Table 2 ijms-24-05265-t002:** The mean total methylation sites in a group of addicted subjects, nicotine-dependent subjects, sports subjects and controls.

Study Groups of Mean Total Methylation Level					F_3,802_ (*p*-Value) [η^2^]
Study Groups	Addicted Subjects n = 274	Nicotine Dependent n = 137	Sports Subjects n = 105	Control n = 290	
M	40.94%	62.84%	65.71%	42.36%	95.185 (<0.000001) [0.262]
SD	15.38%	24.53%	15.01%	15.70%	
−95% CI	39.11%	58.69%	62.81%	40.55%	
+95% CI	42.77%	66.98%	68.62%	44.18%	

M—arithmetic mean, SD—standard deviation, F_3,802_—statistics F ANOVA, η^2^—Eta coefficient.

**Table 3 ijms-24-05265-t003:** Post hoc (Fisher LSD) analysis of dependencies between a group of addicted subjects, nicotine-dependent subjects, sports subjects and controls.

	Addicted Subjects MT = 40.94%	Nicotine Dependent MT = 62.84%	Sports Subjects MT = 65.71%	Control MT = 42.36%
addicted subjects		<0.000001	<0.000001	0.331573
nicotine dependent	<0.000001		0.201627	<0.000001
sports subjects	<0.000001	0.201627		<0.000001
Control	0.331573	<0.000001	<0.000001	

**Table 4 ijms-24-05265-t004:** Primary statistics include 274 addicted subjects, 137 nicotine-dependent subjects, 105 sports subjects and 290 control subjects.

	Addicted Subjects	Nicotine Dependent	Sports Subjects	Control
N	274	137	105	290
Age M(SD)	27.99 (6.20	27.72 (10.18)	23.07 (6.45)	22.17 (4.61)
man/woman	100%/0%	53%/47%	80%/20%	100%/0%

**Table 5 ijms-24-05265-t005:** PCR reaction conditions for the amplification of a 447 bp fragment encompassing 33 CpG sites in the promoter of the *DAT1* gene.

Number of Cycles	PCR Step	Temperature	Time
1	Initial Denaturation Denaturation	94 °C 94 °C	5:00
			0:25
35	Annealing Elongation	61 °C 72 °C	0:25
			0:25
1	Final elongation	72 °C	5:00

## Data Availability

Not applicable.

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
