# Peer review of "DNA Methylation of the Dopamine Transporter DAT1 Gene—Bliss Seekers in the Light of Epigenetics"

_ijms, 2023, doi:10.3390/ijms24065265_

Round 1

Reviewer 1 Report

Research article explains the role of environmental factors in influencing the epigenetic signaling events. DAT1 gene's promoter is hypermethylated in nicotine addicted people as well as in the people doing high sports activities.

I found several loop holes in the paper:

1. What is addicted subjects used in the paper? Why addicted subjects and controls looks very similar in methylation status?

2. Please explain the significance of hypermethylation in nicotine addicted subjects.

3. Discussion of the paper is poorly written, it seems like different people have written different paragraphs, it is not connected

4.Please provide brief introduction about nicotine addiction and hypermethylation in other genes if it is already in the literature

Author Response

Thank you very much for analyzing our Manuscript and for all your comments. We analyzed them thoroughly, and they turned out to be very helpful. We answer each point below while marking corrections in the Manuscript.

  1. What is addicted subjects used in the paper? Why addicted subjects and controls looks very similar in methylation status?

In our study, we recruited a group of patients staying in an addiction treatment centre in closed therapy for at least a year. People recruited to the study group were addicted to psychoactive substances and classified as polytoxic addicts. The control group consisted of mentally healthy people without addictions, also examined by a psychiatrist. Of course, the groups were matched in terms of age and gender.

"Why addicted subjects and controls look very similar in methylation status?" – we ask ourselves this question hundreds of times. We have several hypotheses, but we are rather cautious in concluding because the group was relatively small (for such bold conclusions).

This could be due to the fact that the group of addicted patients was recruited after 3 months of abstinence. This was a necessary condition for reliable psychological testing. Could this time have reduced methylation levels? Were 3 months enough? - we don't know.

Another idea is that the similar level of methylation in the group of addicts and controls was because both groups were relatively "young" and the duration of substance use was shorter than, for example, in nicotine smokers.

We still have many questions, but we know that research must continue.

  1. Please explain the significance of hypermethylation in nicotine addicted subjects.

We suspect the significance of hypermethylation in nicotine-addicted subjects. It was caused by the continuous substance supply several times a day. Alternatively, perhaps nicotine is the factor causing faster methylation than other substances.

Besides, what is more, it may be due to the strictly biological aspect (we have also extended the discussion with this description line: 234-238, page 8.

"The present study provides further evidence that changes in DNA methylation in the regulatory (promoter) regions of addiction-related genes are associated with nicotine addiction, possibly because they can result in altered gene transcription. The rationale is that promoter DNA methylation can directly interfere with binding transcription factors (TFs) to regulatory regions."

  1. Discussion of the paper is poorly written, it seems like different people have written different paragraphs, it is not connected

Thank you for this comment. We've improved the discussion.

4.Please provide brief introduction about nicotine addiction and hypermethylation in other genes if it is already in the literature

A description of other studies has been added to the dissertation – line: 212-223, page 7

"Abdelkhalek et al. measured methylation in the promoter IV of the BDNF gene in smokers compared with healthy controls within 14 days of smoking cessation. Mean methylation was significantly higher in smokers than healthy controls at all time points [33].

Halit et al. assessed the relationship between methylation in the promoters of the APC, NR3C1 and DRD2 genes and its effect on nicotine use. They found a sig-nificant difference in APC2 methylation, comparing the study and control groups. Smokers' methylation rate was about eight times higher than healthy controls. NR3C1 methylation was slightly higher in patients with nicotine addiction com-pared to controls, but the difference was not significant between the two groups (% 95.33 vs. 91.08, p = 0.269). The DRD2 methylation ratio was insignificant between nicotine-addicted patients and healthy controls (P = 0.894)[34]."

Reviewer 2 Report

This is a very interesting manuscript. The phrasing in Lines 122 to 132 is a little unclear. When listing the degree of methylation between groups are they being listed from highest to lowest for each of the CpG islands? Please clarify in the manuscript.

Line 136 refers to table 2. I do not see table 2 in the manuscript, please add. 

Author Response

Dear Reviewer

Thank you very much for analyzing our Manuscript and for all your comments. We analyzed them thoroughly, and they turned out to be very helpful. We answer each point below while marking corrections in the Manuscript.

  1. This is a very interesting manuscript. The phrasing in Lines 122 to 132 is a little unclear. When listing the degree of methylation between groups are they being listed from highest to lowest for each of the CpG islands? Please clarify in the manuscript.

Thank you for this comment. The manuscript description has been updated - line 125-136, page 3

  1. Line 136 refers to table 2. I do not see table 2 in the manuscript, please add. 

We apologize for the typo. It has been corrected, jet is now line 152, page 5.

Round 2

Reviewer 1 Report

Research article explains the role of environmental factors in influencing the epigenetic signaling events. DAT1 gene's promoter is hypermethylated in nicotine addicted people as well as in the people doing high sports activities. Authors have explained the raised concerns. Discussion of the paper is also improved now.